# Transcriptome and Metabolite Profiling of Tomato SGR-Knockout Null Lines Using the CRISPR/Cas9 System

**DOI:** 10.3390/ijms24010109

**Published:** 2022-12-21

**Authors:** Jin Young Kim, Jong Hee Kim, Young Hee Jang, Jihyeon Yu, Sangsu Bae, Me-Sun Kim, Yong-Gu Cho, Yu Jin Jung, Kwon Kyoo Kang

**Affiliations:** 1Division of Horticultural Biotechnology, School of Biotechnology, Hankyong National University, Anseong 17579, Republic of Korea; 2Division of Life Sciences, Korea Polar Research Institute, Incheon 21990, Republic of Korea; 3Department of Biochemistry and Molecular Biology, Department of Biomedical Sciences, Seoul National University College of Medicine, Seoul 03080, Republic of Korea; 4Department of Crop Science, Chungbuk National University, Cheongju 28644, Republic of Korea; 5Institute of Genetic Engineering, Hankyong National University, Anseong 17579, Republic of Korea

**Keywords:** tomato, CRISPR/Cas9, null line, RNA-sequencing, metabolite profiling

## Abstract

Stay-green 1 (SGR1) protein is a critical regulator of chlorophyll degradation and senescence in plant leaves; however, the functions of tomato *SGR1* remain ambiguous. Here, we generated an *SGR1*-knockout (KO) null line via clustered regularly interspaced palindromic repeat (CRISPR)/CRISPR-associated protein 9-mediated gene editing and conducted RNA sequencing and gas chromatography–tandem mass spectrometry analysis to identify the differentially expressed genes (DEGs). *Solanum lycopersicum* SGR1 (SlSGR1) knockout null line clearly showed a turbid brown color with significantly higher chlorophyll and carotenoid levels than those in the wild-type (WT) fruit. Differential gene expression analysis revealed 728 DEGs between WT and sgr#1-6 line, including 263 and 465 downregulated and upregulated genes, respectively, with fold-change >2 and adjusted *p*-value < 0.05. Most of the DEGs have functions related to photosynthesis, chloroplasts, and carotenoid biosynthesis. The strong changes in pigment and carotenoid content resulted in the accumulation of key primary metabolites, such as sucrose and its derivatives (fructose, galactinol, and raffinose), glycolytic intermediates (glucose, glucose-6-phosphate, and fructose-6-phosphate), and tricarboxylic acid cycle intermediates (malate and fumarate) in the leaves and fruit of the SGR-KO null lines. Overall, the SGR1-KO null lines developed here provide new evidence for the mechanisms underlying the roles of SGR1 as well as the molecular pathways involved in photosynthesis, chloroplasts, and carotenoid biosynthesis.

## 1. Introduction

Lycopene is a red carotenoid hydrocarbon found in tomato fruit that is used as a bioactive ingredient to treat chronic diseases and lower the risk of cancer and cardiovascular disease. Numerous studies have attempted to elucidate the pathways involved in lycopene metabolism [1,2,3]. Carotenoid biosynthesis is dependent on isopentenyl diphosphate (*IPP*) and its isomer, dimethylallyl diphosphate [4]. In plastids, four IPP molecules condense into geranylgeranyl pyrophosphate (*GGPP*) molecules. Two molecules of GGPP can then be catalyzed by phytoene synthase 1 (*PSY1*) to form a colorless 15-cis-phytoene molecule, which condenses head-to-head to produce z-carotene and pink prolycopene. Prolycopene is then converted into all-trans-lycopene by carotenoid isomerase. Next, lycopene *β*-cyclase (*LCY-B*) and lycopene ε-cyclase catalyze the cyclization of lycopene to produce β- or α-carotene. Eventually, these substances are broken down into lutein, zeaxanthin, and other carotenoids. Lycopene, a major component of carotenoids in tomatoes, gives color to the fruit and is an important indicator of fruit quality. Therefore, to improve the properties of tomatoes, lycopene accumulation in fruits must be enhanced. Recently, with considerable progress in the modification of the plant genome, regulation of the expression of key genes in the lycopene metabolic pathway has emerged as an effective method by which to increase the lycopene content [5,6,7,8,9,10]. Overexpression of *LCY-B2* or *PSY1* in transgenic plants has been shown to increase the lycopene levels in tomato fruits [11,12]. RNA interference (RNAi) experiments on 9-cis-epoxycarotenoid dioxygenase 1 have reported high accumulation of lycopene and *β*-carotene, resulting in dark red fruits [13]. Some RNAi experiments have been performed by knocking out the green maintenance-related stay-green 1 (*SGR1*) gene. In those plants, lycopene and *β*-carotene in fruits accumulated to levels four and nine times higher than those in the wild-type (WT) plants, respectively. Therefore, suppression of the expression of the *SGR1* gene is attributed to the promotion of *PSY1* activity during fruit ripening [14]. To date, SGR proteins have been reported in *Arabidopsis thaliana* [15,16,17,18,19,20,21], tomato (*Solanum lycopersicum*; [14,22,23,24,25]), rice (*Oryza sativa*; [26,27,28,29,30]), potato (*Ipomoea batatas*; [31,32]), alfalfa (*Medicago truncatula*; [33]) and soybean (*Glycine max* L.; [34]) plants. Most SGR proteins contain a chloroplast transit peptide at the N-terminus, a stay-green domain at the C-terminus, and a cysteine motif [35]. Tomato lycopene has been reported to accumulate via direct interaction with the major carotenoid synthetase, *S. lycopersicum* PSY1 (*SlPSY1*), in targeted mutagenic lines of *SlSGR1* using the clustered regularly interspaced palindromic repeat (CRISPR)/CRISPR-associated protein 9 (Cas9) system [36]. In addition, the CRISPRs/Cas9-induced *SlSGR1* knockout mutant showed drastic color changes in ripened fruits, with significantly higher chlorophyll and carotenoid levels compared to those in WT plants [37].

In this study, we generated a null line via a knockout mutation in the *SGR1* gene using the CRISPR/Cas9 system. The null line showed drastic color changes in ripened fruits and significantly high chlorophyll and carotenoid levels compared to WT plants. We also performed transcriptome and metabolite profiling of the *sgr1* #1-6 line and WT plants. Our results provide new evidence for the mechanisms underlying the roles of *SGR1* as well as the molecular pathways involved in chlorophyll degradation and carotenoid biosynthesis.

## 2. Results

### 2.1. CRISPR/Cas9-Targeted Mutagenesis of SlSGR1

To understand the role of *S. lycopersicum SGR1* (*SlSGR1*) in the ripening of tomato fruits, we generated tomato *SlSGR1* mutants using the CRISPR/Cas9 system. *SlSGR1* encodes a chloroplast signal peptide, a stay-green domain, and a C-terminal cysteine-rich motif, which consists of four exons (Figure 1A). To obtain *SlSGR1* mutants, two guide RNAs (gRNAs), gRNA1 and gRNA2, were designed to target the third and fourth exon, respectively (Figure 1A; Appendix A). Two *SlSGR1* CRISPR/Cas9 constructs containing each gRNA were used to transform the tomato inbred line K19, which is widely used as a crossing parent for F_1_ variety studies of pink tomatoes in the Breeding Lab of Hankyong University (Figure 1B,C). 

We generated 87 and 69 T_0_ transformants for the constructs containing gRNA1 and gRNA2, respectively, and confirmed T_0_ transformants harboring target mutations for both constructs via polymerase chain reaction (PCR)-based genotyping (Figure 2A; Table 1). Deep-sequencing analysis of these variants revealed that chimeric, biallelic, heterozygous, and homozygous *SlSGR1* mutations were present in the T_0_ generation (Figure 2B). Most mutants exhibited 1–19 bp in-frame deletions at the target site, but some mutants showed in-frame additions of 1 bp.

### 2.2. Selection and Characterization of sgr1 Null Lines

T_1_ seeds were harvested from 18 edited null lines (Figure 3A,B). For further analysis, the *sgr1* #1−6 (–19/–19) and *sgr1* #2−4(–5/–5) lines showing large deletions and homologous mutations were selected. Transgene-free mutant lines were screened via PCR-based genotyping using neomycin phosphotransferase II, which is present in the T-DNA region (Figure 3C). We found that the *sgr1* #1−6 and *sgr1* #2−4 lines were segregated in a 3:1 (present: absent) ratio, suggesting that T-DNA was introduced with a single copy into the tomato genome (Table 2).

Transgene-free plants were observed from the *sgr1* #1−6 and *sgr1* #2−4 lines with a proportion of 20%. These results indicate that transgene-free homozygous mutants can be easily obtained in the T1 generation, as the inheritance of T-DNA and the edited gene are relatively independent. We also checked whether these lines had other T-DNA vector sequences apart from the expected T-DNA insertion. This information is important for field cultivation of non-genetically modified plants. Whole-genome sequencing data for *sgr1* #1−6 and *sgr1* #2−4 were obtained via Illumina sequencing. In total, 412 million paired-end reads of raw data were produced. After quality trimming, the average genome coverage was 20×. The reads were mapped against the sequence of the transformation vector to validate the assumption of a single T-DNA insertion based on segregation analysis. The coverage of T-DNA reads was 20×, like the average genome coverage, thus confirming a single-copy locus. For transgenic lines, whole-genome data were mapped not only against the T-DNA but also against a 400-bp vector backbone region. However, the data obtained from the *sgr1* #1−6 and *sgr1* #2−4 lines were mapped only with the intrinsic U6 promoter region, which was isolated from the tomato genome. We also investigated potential off-target mutations in the T_2_ generation using the two selected lines (Appendix A). Six potential off-target sites, including four mismatched bases, were investigated using Cas-OFFinder (http://www.rgenome.net/cas-offinder/ accessed on 7 May 2021) (Appendix A). PCR products obtained from *sgr1* #1−6 and *sgr1* #2−4 lines were sequenced. No mutations were observed in any of the 10 potential off-target sites, indicating that the mutagenesis of the predicted site occurred with high specificity (Appendix A).

### 2.3. Carotenoid Profiles in the Leaves and Fruits of slsgr1 Mutant Lines

Tomato fruits and leaves from *sgr1* #1−6 and *sgr1* #2−4 lines at Br+7 ripening were sampled for carotenoid profiling via high-performance liquid chromatography (HPLC) analysis. The gene-edited lines, *sgr1* #1−6 and *sgr1* #2−4, showed a reduction in total leaf carotenoid levels, with a marked decrease in lutein levels and a marked increase in violaxanthin and zeaxanthin levels, compared to those in WT plants (Appendix A). In the fruit, the levels of lycopene and *β*-carotene in *sgr1* #1−6 and *sgr1* #2−4 lines were much higher than those in the WT plants (Table 3). 

In particular, the lycopene levels of *sgr1* #1−6 were the highest. Chlorophyll levels were increased in *sgr1* #1−6 and *sgr1* #2−4 lines compared to those in WT plants (except for a slight increase in chlorophyll b levels in *sgr1* #2−4 line) (Table 4).

### 2.4. Transcriptome Analysis of sgr#1−6 Mutant and WT Plants

Six samples from three biological replicates (three *sgr1* #1−6 vs. three WT) were used for RNA sequencing analysis at the breaker (Br) stage. The number of reads per sample ranged from 44,386,446 to 74,488,546 among the six sequenced RNA samples (Appendix A). Principal component analysis of *sgr1* #1−6 and WT libraries was used to determine data clustering based on *SGR1* expression. All biological replicates of the *sgr1* #1−6 and WT plants were distributed in two distinct groups (Figure 4A). Differential gene expression analysis revealed 728 differentially expressed genes (DEGs) between WT and *sgr1* #1−6 line, including 263 downregulated and 465 upregulated genes, for which the fold-change was >2 and the adjusted *p*-value was <0.05. (Figure 4B). According to gene ontology (GO) enrichment analysis (adjusted *p* < 0.1; Figure 4C), several DEGs were associated with the following GO terms: fruit ripening (GO:0009835), sterol metabolic process (GO:0016125), cytoplasm (GO:0005737), oxidation–reduction process (GO:0055114), and extracellular region (GO:0005766) (Table 5). DEGs were also analyzed using STRING (version 11.5) to construct a protein–protein interaction (PPI) network (Figure 4D); PSY1 (degree = 21; adjusted *p* = 1.82 × 10^−8^) and protein kinase (Solyc05g018300; degree = 25; adjusted *p* = 1.65 × 10^−8^) were the top one-degree proteins (Figure 4D). To verify the RNA sequencing data, nine DEGs (*Fab G*, *AP2a*, *DDTFR8*, *RIN*, *LOXB*, *ERF*-*D2*, *LOXC*, ACC oxidase (*ACO*-*1*, and *ACO6*) associated with the fruit ripening were selected, and their mRNA expression levels were verified in biologically replicated *sgr1* #1−6 lines via reverse transcription-quantitative PCR (RT-qPCR) analysis. Figure 5 shows the fold-changes in these genes between the *sgr1* #1−6 line (KO) and WT plants at the fruit ripening stage.

### 2.5. Metabolite Profiling of the sgr1 #1−6 Null Lines

Potent changes in pigment and carotenoid levels led us to investigate the impact of these changes on other metabolic pathways. Gas chromatography–mass spectrometry (GC-MS) metabolite profiling showed significant changes in the levels of sucrose and its derivatives (fructose, galactinol, and raffinose), glycolytic intermediates (glucose, glucose-6-phosphate [G6P], and fructose-6-phosphate [Fru6P]), and tricarboxylic acid (TCA) cycle intermediates (malate and fumarate) in the leaves and fruits of *sgr1* #1−6 and WT plants (Figure 6; Appendix A.). These changes were reflected in the levels of G6P-derived compounds (trehalose, maltotriose, maltose, *myo*-inositol, and erythritol) and amino acids derived from glycerate (O-acetylserine), pyruvate (valine, alanine, and leucine), shikimate (phenylalanine and tryptophan), malate (aspartic acid, asparagine, *β-*alanine, and methionine), and 2-oxoglutarate (glutamic acid, glutamine, GABA, and ornithine) (Figure 6).

## 3. Discussion

Carotenoids are essential pigments in photosynthetic organisms, and their accumulation produces color in flowers and fruits. The mechanism of carotenoid accumulation in tomatoes is related to the expression of genes encoding carotenoid biosynthesis enzymes during fruit ripening [38]. Lycopene is a bright-red carotenoid hydrocarbon found in tomatoes. It is a bioactive ingredient that is used to treat chronic diseases and lower the risk of cancer and cardiovascular diseases. So far, researchers have overexpressed several genes and pathways related to lycopene metabolism to increase lycopene levels in tomato fruit [1,2,3]. In this study, we generated stable *slsgr1* mutants using CRISPR/Cas9 gene-editing technology to understand the role of *SlSGR1* in tomato fruit ripening (Figure 1). The fruit of *slsgr1* mutants changed from a green to a red color during the ripening process (Figure 3b). Our results indicate that the fruit phenotypes of the SlSGR1::RNAi and SISGR1 knockout lines are similar [14,37,39]. Transgene-free mutant lines were screened via PCR-based genotyping using neomycin phosphotransferase II, which is present in the T-DNA region (Figure 3c). We found that the *sgr1* #1−6 and *sgr1* #2−4 lines were segregated in a 3:1 (present: absent) ratio, suggesting that T-DNA was introduced with a single copy into the tomato genome (Table 2). We also checked whether these lines still had other T-DNA vector sequences besides the expected T-DNA insertion. To this end, we generated whole-genome sequencing data for *sgr1* #1−6 and *sgr1* #2−4 obtained through Illumina sequencing. For the transgenic lines, whole-genome data were mapped, not only to the T-DNA, but also to the 400-bp vector backbone region, which mapped only to the unique U6 promoter region isolated from the tomato genome. Therefore, no fragments were inserted from the outside into the genomes of *sgr1* #1−6 and *sgr1* #2−4 lines (Appendix A). 

The *sgr1* #1−6 and *sgr1* #2−4 lines showed a reduction in total leaf carotenoid levels, with a marked decrease in lutein and a marked increase in violaxanthin and zeaxanthin levels compared to those in WT plants (Appendix A). In the fruit, the levels of lycopene and *β-*carotene in *sgr1* #1−6 and *sgr1* #2−4 lines were much higher than those in the WT plants (Table 2). Therefore, the color change of the fruit was positively correlated with these parameters. Our results indicate that *SlSGR1* critically affects the color changes in ripening fruits via chlorophyll degradation and carotenoid biosynthesis. Many studies have investigated the influence of the accumulation of carotenoids and the decomposition of chlorophyll on color change during tomato ripening. Based on the results of recent studies, non-coding RNAs and numerous transcription factors have been suggested to function as regulators. One study reported that red coloration was delayed during the ripening of tomato fruit via RNAi experiments using *SlWRKY16*, *SlWRKY 17*, *SlWRKY53*, *SlWRKY54*, and *WRKY* transcription factors [40]. Another study reported that the *FveMYB10* transcription factor restored the red pigment in an experiment using the yellow fruit of *Fragaria vesca* [41]. In the present study, fruits obtained from the *slsgr1* #1 null line, which lacked *SlSGR1* function, were initially green and turned reddish-brown in color at the ripening stage. In particular, the change from pink to reddish-brown color in fruits may affect the expression of genes required for fruit ripening. To elucidate the molecular mechanism of color change in the ripening process of tomatoes, RNA-sequencing analysis was performed using the fruits of the *sisgr1* #1 line and the BR+7 stage of WT plants (Figure 4). Differences in the expression levels of *Fab G*, *AP2a*, *DDTFR8*, *RIN*, *LOXB*, *ERF-D2*, *LOXC*, *ACO1*, and *ACO6* genes were related to photosynthesis (Figure 5). Upregulation of these genes in the *slsgr1* mutant may be due to aberrant chloroplasts and the effects of light capture [42,43]. Therefore, further studies are necessary to determine the effects of specific genes on *slsgr1* mutant fruit quality and chlorophyll metabolism. In this study, when the *SlSGR1* gene was knocked out using CRISPR/Cas9 in pink tomatoes, not only did the color of the fruit change, but the lycopene content was also significantly improved, suggesting that the expression of various genes (nine DEGs: *Fab G*, *AP2a*, *DDTFR8*, *RIN*, *LOXB*, *ERF*-*D2*, *LOXC*, *ACO1*, and *ACO6*) was affected. In addition, GC-MS metabolite profiling was performed to determine the substances that were altered in the metabolite pathway due to carotenoid and pigment changes in the *slsgr1* #1−6 line (Appendix A). In the leaves and fruits of *sgr1* #1−6 and WT plants, significant differences were observed in the levels of sucrose and its derivatives (fructose, galactinol, and raffinose), glycolysis intermediates (glucose, G6P, and Fru6P), and TCA cycle intermediates (malate and fumarate) (Figure 6). Accumulation of sugars and derivatives (raffinose, galactinol, *myo*-inositol, and trehalose) and amino acids (Val, Asp, Asn, Thr, Glu, Gln, and Ala) in fruits has been reported to confer tolerance to chilling injury and resistance to pathogens and several post-harvest stress conditions [44,45,46,47]. Therefore, the accumulation of these metabolites is expected to confer beneficial post-harvest characteristics, such as improved fruit quality and extended shelf-life, to tomatoes. The transcriptomic and metabolite profiles of SGR1-KO lines presented here provide new evidence for the mechanisms underlying the roles of SGR1 as well as the molecular pathways involved in chlorophyll degradation and carotenoid biosynthesis.

## 4. Materials and Methods

### 4.1. Plant Materials and Growth Conditions

Tomato (*S. lycopersicum* L.) was obtained using a K19 inbred line (pink color), which was grown in the breeding lab of Hankyong University. All plants were grown under the same greenhouse conditions. The ripening stages were divided into MG, Br+0, Br+4, Br+7, and Br+10. The samples were collected from fruits and leaves, immediately frozen in liquid nitrogen, and stored at –80 °C until analysis. After harvesting the mature seeds from transgenic T0 plants, they were dried and stored in a refrigerator.

### 4.2. Plasmid Construction and Genetic Transformation of Tomato

Target sites and single guide RNAs (sgRNAs) for the third and fifth exons of *SlSGR1* (Solyc08g080090) adjacent to a protospacer-adjacent motif were amplified using specific primer sets (Appendix A). The CRISPR/Cas9 vector was constructed by selecting three target sites in the *SlSGR1* sequence using the CRISPR RGEN program (http://www.rgenome.net/, accessed on 20 April 2022) (Table 1). A 20-nt sgRNA scaffold sequence was synthesized by Bioneer Co., Ltd. (Daejeon, Republic of Korea) and the dimer was cloned into an *Aar*I-digested 35s-p:pKAtC binary vector. The constructs thus obtained were transformed into tomato cotyledons using *Agrobacterium tumefaciens* strain EHA105. The transformed cotyledons were selected using 50 mg/L kanamycin and confirmed via PCR analysis. To verify the target site mutations, PCR amplicons were subjected to MiniSeq paired-end read sequencing (Illumina, San Diego, CA, USA) and analyzed using a Cas-Analyzer (https://www.rgeno me.net/cas-analyzer/ accessed on 21 July 2020).

### 4.3. Mutation Analysis of Transgenic Lines

Total DNA extraction was performed by crushing 0.3 g of tomato leaves under liquid nitrogen with an electric drill in a 1.5-mL Eppendorf tube. A volume of 700 μL extraction buffer (DNA extract kit, Bio Co., Tajeon, Republic of Korea) was added to each tube and incubated for 20 min at 65 °C. A volume of 800 μL chloroform-isoamyl alcohol (24:1) was added to each tube and centrifuged at 12,000 rpm for 7 min; this step was repeated twice. All other tests were performed according to the method specified in the DNA extract kit (DNA extract kit, Bio Co., Tajeon, Republic of Korea). The extracted genomic DNA was used as a template to amplify the relevant fragments from each target gene using primers (Appendix A) flanking the target sites. The standard PCR conditions were as follows: 94 °C for 7 min, 30 cycles of 94 °C for 30 s, 58 °C for 30 s, 72 °C for 1 min, and 72 °C for 7 min. PCR products were directly sequenced via deep sequencing using internal primers (Appendix A) to identify mutations. The mutation rate for each target was calculated as the ratio of the number of transgenic plants edited in each target to the total number of transgenic plants obtained.

### 4.4. Determination of Chlorophyll and Carotenoid Levels

Carotenoids in tomato leaves and fruits were extracted with a 0.01% solution of butylated hydroxytoluene in acetone and analyzed using an Agilent 1260 HPLC system (Hewlett-Packard, Waldbronn, Germany). Carotenoids were quantified using an external calibration method. *β-*Carotene, lutein, violaxanthin, and zeaxanthin standards were purchased from Sigma Co (Seoul, Republic of Korea). All extraction procedures were performed under dim light conditions. Carotenoid content was calculated as µg g^−1^ dry weight of leaf and fruit tissue.

### 4.5. RNA Extraction, Library Preparation and RNA-Seq

Total RNA samples were prepared from WT and *slsgr1* null mutant fruits at BR (three replicates per sample) using TRIzol reagent. RNA integrity was assessed via agarose gel electrophoresis. RNA-seq libraries were constructed and sequenced on an Illumina HiSeq PE150 platform with 150-bp paired-end reads (Novogene, Tianjin, China). The generated raw data had a sequencing depth of at least 4 Gb. All raw reads have been deposited at THERAGEN Bio Co. (http://www.ftp.theragenbio.com accessed on 20 April 2022) under accession number TBD211714_14868_20220704. Clean reads were checked by basic quality statistics using FastQC software (http://www.bioinformatics.babraham.ac.uk/projects/fastqc accessed on 20 April 2022). Reads with low-quality and adapter sequences were removed, after which the FastQC step was run again. The reads were mapped to the tomato reference genome (version SL2.40) using TopHat (version 1.4.6, http://ccb.jhu.edu/software/tophat/index.shtml accessed on 20 April 2022) and fragments were assigned to genes using FeatureCounts (version 2.0.14, http://subread.sourceforge.net/ accessed on 20 April 2022) [48,49]. The FPKM (fragments per kilobase of transcript sequence per millions base pairs sequenced) of each gene was then calculated based on the gene length and reads, with 425 counts mapped to this gene. Differential gene expression was determined using the criteria of fold-change >2, which was analyzed using the DESeq2 package (https://bioconductor.org/packages/release/bioc/html/DESeq2.html accessed on 20 April 2022) and adjusted *p*-value. The *p*-values obtained were adjusted using the Benjamini and Hochberg approach to control the false-discovery rate. GO enrichment analysis of DEGs was conducted using ClusterProfiler (v3.10.1). PPI networks were generated and analyzed using STRING v11.5. RT-qPCR assays were performed according to the standard protocol established in our laboratory.

### 4.6. qRT-PCR Analysis 

Total RNA was extracted from tomatoes using a RNeasy Plant Mini Kit (Qiagen, www.qiagen.com accessed on accessed on 23 July 2022), and single-stranded cDNA was synthesized with random oligonucleotides using a reverse transcription system (Takara, www.takara-bio.com accessed on 23 July 2022). To analyze the expression levels of DEGs, qRT-PCR was performed using cDNA, gene-specific primers, and SYBR Green Real-time PCR Master Mix (Toyobo, http://www.toyobo.co.kr/ accessed on 23 July 2022), according to the manufacturer’s instructions. The sequences of gene-specific primers used for amplification are listed in Appendix A. The *SlActin* gene was used as an internal standard, and relative gene expression levels were calculated using the comparative Ct method.

### 4.7. Resequencing Analysis

Total RNA was extracted from the Genomic DNA isolated from the leaf material of the T1 null lines (transgene-free edited homozygous mutant lines) and subjected to Illumina sequencing. Whole-genome shotgun libraries were subjected to paired-end sequencing analysis on TERAZEN. Sequence reads of the samples were aligned to a reference consisting of the tomato reference genome (https://www.ncbi.nlm.nih.gov/assembly/313038/ accessed on 10 Jun 2022) and the vector sequence with pKAtC. Read depth was calculated with the command samtools depth using only uniquely aligned reads with a mapping quality of 20 or greater and plotted with standard R functions (R Core Team 2015).

### 4.8. Metabolite Profile Analysis 

Polar metabolites were extracted as previously described [50]. Metabolites were extracted from powdered tissues (100 mg) by adding 1 mL of 2.5:1:1 (*v*/*v*/*v*) methanol:water:chloroform. For metabolites, chromatograms and mass spectra were evaluated as described previously [50]. ChromaTOF software was used to support the peak results prior to quantitative analysis and automatic deconvolution of the reference mass spectra. In-house libraries of NIST and standard chemicals were used for compound identification. The calculations used to quantify the concentrations of all analytes were based on the ratio of the peak area of each compound to that of the internal standard.

## Figures and Tables

**Figure 1 ijms-24-00109-f001:**
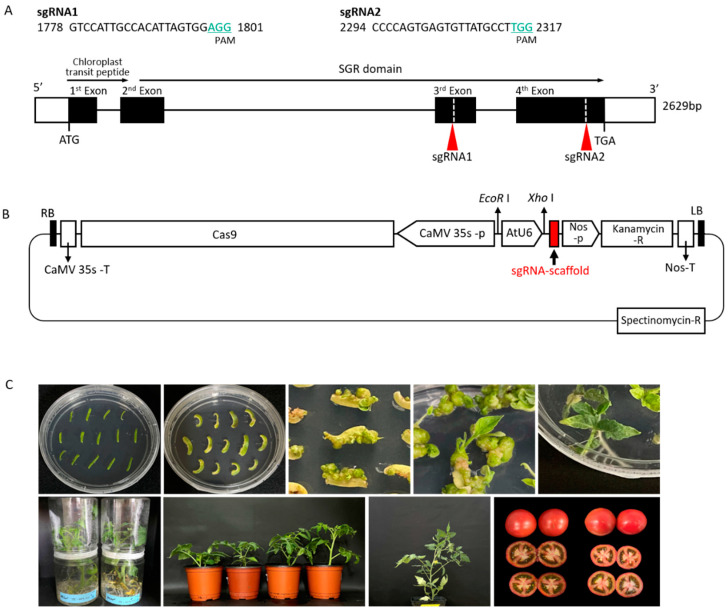
Clustered regularly interspaced palindromic repeat (CRISPR)/CRISPR-associated protein 9 (Cas9)-induced *Solanum lycopersicum* stay-green 1 (*SlSGR1*) gene editing. (**A**) Schematic of the single guide RNA (sgRNA) target site in the genomic region of *SlSGR1*. (**B**) Ti-plasmid vector construction of sgRNA region for gene editing in tomato. (**C**) Generation of edited lines of *SlSGR1* gene using the CRISPR/Cas9 system. Red arrows indicate the sgRNA1 and sgRNA2 regions.

**Figure 2 ijms-24-00109-f002:**
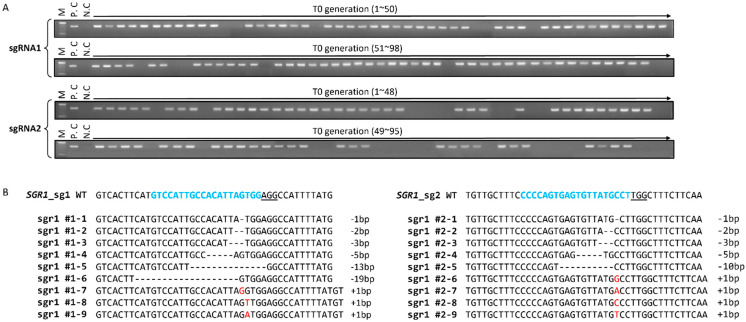
CRISPR/Cas9-induced mutation in *SlSGR1* in tomato. (**A**) Confirmation of transformed plants via agarose gel electrophoresis with polymerase chain reaction (PCR) products amplified using the Ti-plasmid region from putative transgenic plants. (**B**) Deep-sequencing analysis of edited plants. Target DNA sequences of *SGR1*_sg1 and *SGR1*_sg2 are shown with the protospacer-adjacent motif (PAM) region in blue color on top of the aligned sequences. Deletions are indicated by dashes and insertions are indicated by red color. Indel sizes are shown on the right (+, insertion; −, deletion).

**Figure 3 ijms-24-00109-f003:**
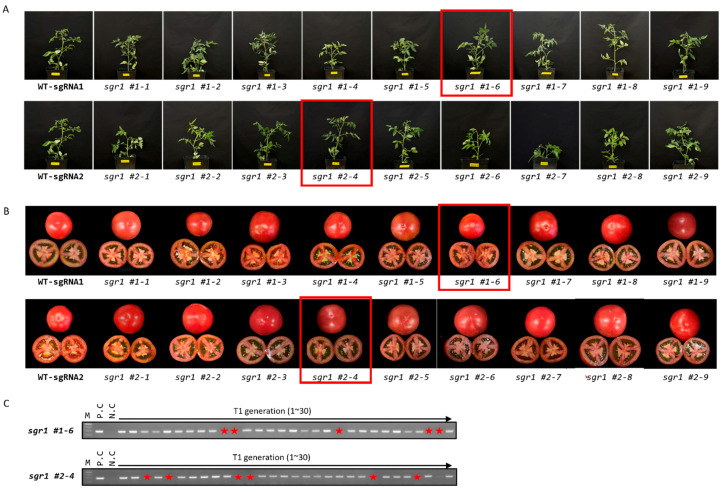
Phenotypes based on editing of the *SlSGR1* gene in tomatoes. (**A**) Phenotypes of edited lines. (**B**) Fruit shapes in T_1_ generation. (**C**) Selection of null segregant lines in T_1_ generation via PCR analysis. Lanes with no band indicate T-DNA was removed.

**Figure 4 ijms-24-00109-f004:**
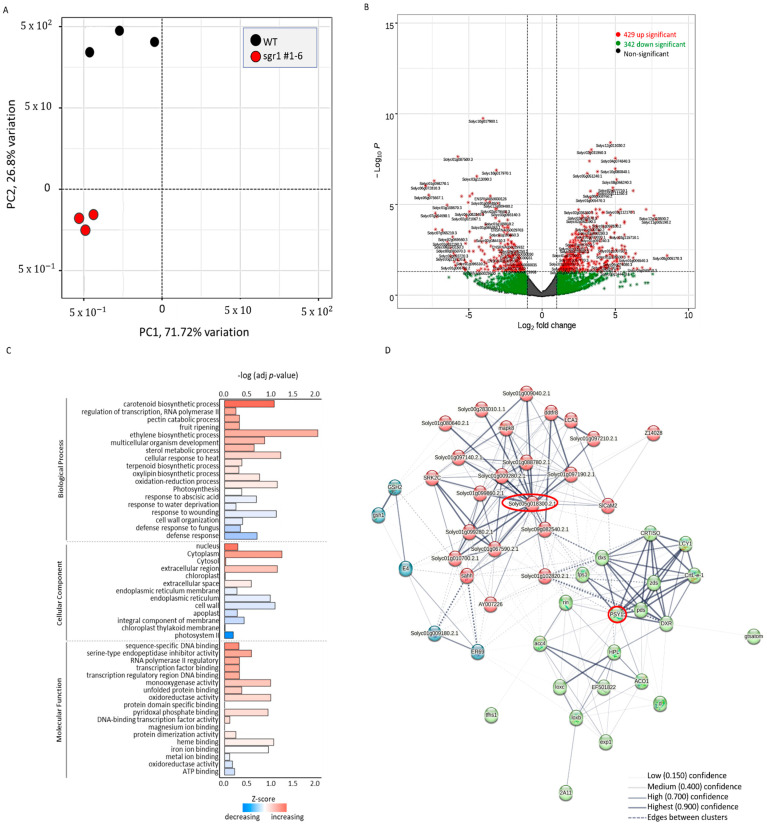
Analysis of RNA sequencing data. (**A**) Principal component analysis plot. (**B**) Volcano plot showing the differentially expressed genes (DEGs) between wild-type (WT) and sgr1-mutant plants. (**C**) Gene ontology (GO) terms associated with DEGs in biological process (BP), cell component (CC), and molecular function (MF) ontologies. (**D**) Protein–protein interaction network constructed using the top 50 DEGs.

**Figure 5 ijms-24-00109-f005:**
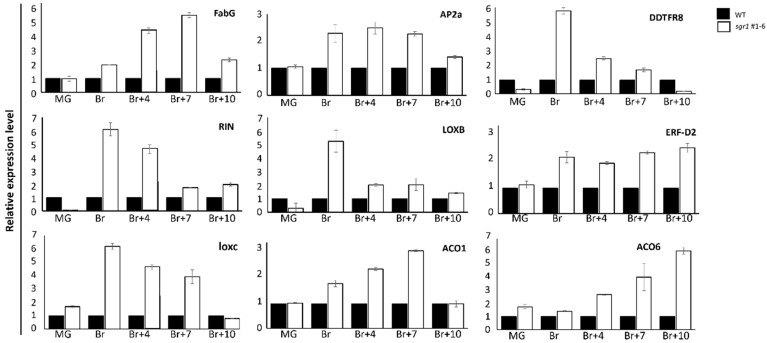
Relative gene expression levels of *sgr1* #1−6 null lines and WT plants via RT-qPCR analysis. Error bars indicate mean ± standard deviation of three biological replicates, each measured in triplicate.

**Figure 6 ijms-24-00109-f006:**
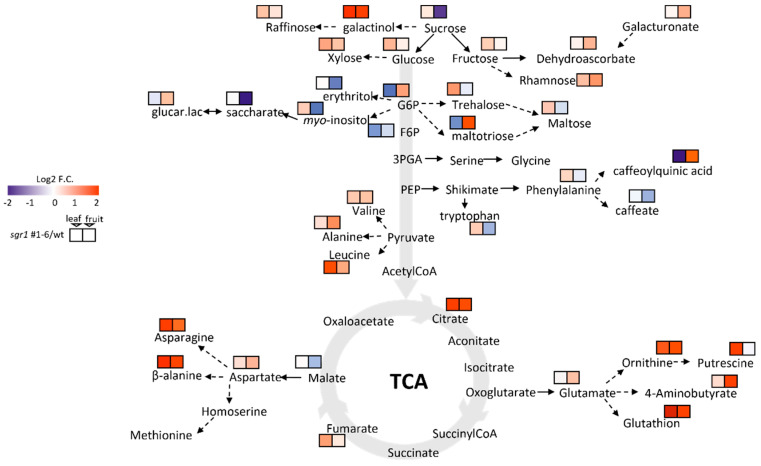
Metabolite profiling of sucrose and its derivatives, glycolytic intermediates, and TCA cycle intermediates in the leaves and fruits as a result of *sgr1* knockout in tomato. Primary metabolite profiling of Br+7 stage fruits and leaves in *sgr1* #1−6 and WT plants. Ratios between *sgr1* #1−6 and WT plants are shown. A non-paired two-tailed Student’s *t*−test was used to compare each *sgr1* #1−6 line with its WT plant (*p* < 0.05; *n* = 5).

**Table 1 ijms-24-00109-t001:** Frequencies of transgenic plants and genotypes of the target gene region based on genome editing of *Solanum lycopersicum* stay-green 1 (*SlSGR1*) gene using the clustered regularly interspaced palindromic repeat (CRISPR)/CRISPR-associated protein 9 (Cas9) system.

Target Region	No. of Regenerated Plants	No. of Transgenic Plants	No. of Edited Plants	Genotype			
				Homoallelic	Heteroallelic	Biallelic	Multiallelic
*SlSGR1*-sg 1	98	87	37	8	6	11	12
*SlSGR1*-sg 2	95	69	31	4	7	9	11

**Table 2 ijms-24-00109-t002:** Chi-square analysis for resistant and susceptible strains from T1 generation of SGR1 edited plants.

T1 Generation	Total Seeds	No. of Resistance	No. of Susceptible	X^2^-Test	*p*-Value
*sgr1* #1−6	40	31	9	0.72	0.5 < *p* < 0.2
*sgr1* #2−4	39	30	9	0.78	0.5 < *p* < 0.2

**Table 3 ijms-24-00109-t003:** Determination of carotenoid levels in *sgr1* #1−6, *sgr1* #2−4, and wild-type (WT) plant fruits at Br+7.

Lines	Lutein	*β*-Carotene	Lycopene	Others	Total
	Mean ± SD *	Mean ± SD *	Mean ± SD *	Mean ± SD *	Mean ± SD *
WT	7.3 ± 1.9	18.1 ± 2.9	225.9 ± 25.6	142.5 ± 11.0	393.8 ±41.5
*sgr1* #1−6	4.3 ± 0.1	48.0 ± 6.6	999.5 ± 3.5	362.4 ± 15.3	1414.2 ± 25.5
*sgr1* #2−4	5.8 ± 0.1	30.8 ± 2.2	809.9 ± 8.8	317.4 ± 8.1	1263.9 ± 19.2

* Data are expressed as the mean (the average value of content for dry weight) and standard deviation (SD) of three independent experiments. Carotenoid content was calculated as µg g^−1^ dry weight of fruit tissue.

**Table 4 ijms-24-00109-t004:** Determination of chlorophyll levels in sgr1 #1-6, sgr1 #2-4, and wild-type (WT) plants.

Samples	Chlorophyll a	Chlorophyll b
	Mean ± SD *	Mean ± SD *
WT	5670.1 ± 226.3	2302.4 ± 100.6
*sgr1* #1−6	7911.5 ± 113.2	3399.3 ± 63.1
*sgr1* #2−4	7491.1 ± 340.0	2934.9 ± 115.4

* Data are expressed as the mean (the average value of content for dry weight) and standard deviation (SD) of three independent experiments. Chlorophyll content was calculated as µg g^−1^ dry weight of leaf tissue.

**Table 5 ijms-24-00109-t005:** Top five gene ontology (GO) terms enriched in SGR1-related differentially expressed genes (DEGs).

Ontology *	GO Accession	GO Term	Gene Count	Gene IDs	Adjusted *p*-Value
BP	GO:0009835	Fruit ripening	15	Solyc01g095080.3, Solyc02g091990.3, Solyc03g093610.1, Solyc03g118290.3, Solyc05g005560.3, Solyc05g050010.3, Solyc07g049530.3, Solyc07g049550.3, Solyc07g056570.1, Solyc07g064190.2, Solyc08g005610.3, Solyc09g089580.3, Solyc09g092480.2, Solyc10g080210.2, Solyc12g005940.2	1.10 × 10^−7^
BP	GO:0016125	Sterol metabolic process	10	Solyc01g109140.3, Solyc02g065750.2, Solyc02g070580.1, Solyc02g089160.3, Solyc04g078900.3, Solyc04g079730.1, Solyc07g049690.3, Solyc08g005610.3, Solyc10g007960.1, Solyc11g069800.1	6.70 × 10^−5^
CC	GO:0005737	Cytoplasm	39	Solyc01g099190.3, Solyc01g099760.3, Solyc01g099770.3, Solyc01g101170.3, Solyc01g101180.3, Solyc01g103390.3, Solyc01g111450.3, Solyc02g080790.3, Solyc03g111720.3, Solyc04g071260.3, Solyc04g082030.1, Solyc05g007940.3, Solyc05g023800.3, Solyc05g051750.3, Solyc06g005060.3, Solyc06g005260.3, Solyc06g009970.3, Solyc06g059885.1, Solyc06g073390.3, Solyc06g074350.3, Solyc06g076020.3, Solyc06g076570.2, Solyc06g083230.3, Solyc07g065840.2, Solyc08g014000.3, Solyc08g043170.3, Solyc08g080650.2, Solyc08g082820.3, Solyc09g007910.3, Solyc09g009390.3, Solyc09g092480.2, Solyc10g080500.2, Solyc10g085280.2, Solyc10g085870.1, Solyc10g086220.2, Solyc10g086410.3, Solyc12g006470.2, Solyc12g035890.2, Solyc12g098940.2	6.30 × 10^−5^
BP	GO:0055114	Oxidation–reduction process	11	Solyc01g109140.3, Solyc02g065750.2, Solyc02g070580.1, Solyc02g089160.3, Solyc04g078900.3, Solyc04g079730.1, Solyc07g049690.3, Solyc08g005610.3, Solyc10g007960.1, Solyc10g086500.1, Solyc11g069800.1	7.90 × 10^−4^
CC	GO:0005576	Extracellular region	21	Solyc01g008710.3, Solyc01g009590.3, Solyc01g067740.3, Solyc01g097240.3, Solyc02g079500.3, Solyc02g089350.3, Solyc02g093580.3, Solyc03g020060.3, Solyc03g123620.3, Solyc05g007940.3, Solyc05g007950.3, Solyc06g064520.3, Solyc06g068520.3, Solyc07g064190.2, Solyc08g080630.3, Solyc08g080650.2, Solyc09g084460.3, Solyc10g055810.2, Solyc11g011210.2, Solyc11g066130.1, Solyc11g066390.2	7.90 × 10^−4^

* CC, cellular component; BP, biological component.

## Data Availability

Not applicable.

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
