# Peer review of "Transcriptome and Metabolite Profiling of Tomato SGR-Knockout Null Lines Using the CRISPR/Cas9 System"

_ijms, 2022, doi:10.3390/ijms24010109_

Round 1

Reviewer 1 Report

The authors describe the generation of two different CRISPIR-Cas9 mediated mutations in the SGR1 gene in Tomato. A strong phenotype is observed in the tomato fruits with increase in red/brown coloration, indicating a potential increase in lycopene accumulation. Multiple molecular analyses were done that compare one of these transgenic lines with the wildtype tomato line. These analyses include carotenoid and chlorophyll quantification, transcriptomics and metabolomics. In each case, significant differences in molecular abundances were observed between mutant lines and wildtype.

Major Comments

Overall, I see no major flaws in the transcriptome analysis. Likewise, I do not see major flaws in the GC-MS metabolite profiling. However, I also see no useful conclusions drawn from these analyses. In both cases a list of differentially accumulated transcripts or metabolites were found and no further conclusions were drawn. Similarly, a STRING protein-protein interaction network was generated but nothing was said about this.

My biggest concern is that this manuscript has little or no impact. This manuscript cites Zeng et at. 2022 who have recently reported SGR1 knockout along with transcriptome analysis. It is not clear how this study adds knowledge or resources to the field.

The authors claim a null mutation.

I think the sequencing data is convincing and shows a homozygous deletion (likely also a frameshift) mutation. Also, the phenotype is quite dramatic so I have no problem with claims that the induced mutation are effecting the phenotype. However, the authors did not conclusively prove a null mutation. It would be good to see RNA-seq quantification for SGR1 in the mutant line. This could show potentially decreased expression. More significantly, the read depth of these experiments may be sufficient to show a truncated transcript. This data may be enough to prove a null mutation. If not, the authors should not make this claim.

Overall, the manuscript was difficult to read for several reasons:

1)    The manuscript would benefit from further editing. grammatical errors make it difficult to understand certain passages (two examples below, but many exist in the text)

2)    The introduction could be better organized. The basic carotenoid biosynthesis pathway is described and this would benefit from a diagram of the pathway.

3)    Several different knockdown and knockout studies are cited in the intro but the text does not bring the results of these studies together and it is not clear how the gene of interest (SGR1) relates to the existing knowledge cited.

Two examples of grammatical errors that make the text unclear:

Line 52:“Overexpression of the lycopene β-cyclase 2 (LCY-B2) gene or the phytoene synthase 1 gene (PSY1) in transgenic plants have been increased lycopene levels in tomato fruits”

Maybe should read: “Overexpression of the lycopene β-cyclase 2 (LCY-B2) gene or the phytoene synthase 1 gene (PSY1) in transgenic plants has been show to increased lycopene levels in tomato fruits”

Line 56: “In addition, the RNAi experiment of the green maintenance-related SGR1 gene, the contents of lycopene and β-carotene were accumulated 4 times and 9 times in fruit, respectively”

Maybe should read: “In addition, RNAi experiments have knocked down the green maintenance-related SGR1 gene. In these plants, lycopene and β-carotene in fruit accumulated to levels 4 and 9 times higher than wildtype, respectively”.

Line 85: “and confirmed T0 transformants harboring target mutations for all two constructs using PCR-based genotyping (Figure 2A, Table 1).”

This sentence confuses the reader. The phrase “for all two constructs” does not make sense. Also, the term PCR-based genotyping and referring to Fig2A seems to indicate that targeted mutations were confirmed via the gels in Fig2A. If I understand correctly, the authors do confirm target mutations but this is done through deep sequencing. This is not explained clearly in the results or methods section.

Please clarify if the T0 or T1 lines (lacking the T-DNA) were used for all further analysis.

Line 131

Chlorophyll content increased in sgr1 #1-6 and sgr1 #2-4 lines compared to WT (except for a  slight decrease in chlorophyll b in sgr1 #2-4) (Table 3)”

Table 3 shows a slight increase in IIb, not a decrease

Line 148

Principal component analysis of sgr1 #1-6 and WT libraries was used to determine data clustering based on SGR1 expression. All biological replicates of sgr1 #1-6 and WT plants were distributed in two distinct groups (Figure 4A).”

Figure 4A only seems to show one replicate from each group. This plot should show a point for each sample

GO enrichment analysis was carried out on DEGs. The authors should report details on how this analysis was carried out in the methods section.

Figure 4 legend seems to indicate annotations for significance levels in the GO analysis. However, the figure contains none of these annotations.

Table 5: It seems that all but one of the GO categories listed in this table are not significant (p-values > 0.05)

Minor comments

Please double check references. As an example:

These results were similar to the fruit phenotypes generated in the SlSGR1-RNAi tomato line as well as in the SISGR1 knockout line using the CRISPR-Cas9 gene editing 205 technique [14, 37, 39]”

It seems like this should be referencing [36], Zeng et al., 2022.

Line 301 Typo:

All raw reads are deposited at All raw reads are deposited at TeRaGene”

Line 325:

For metabolites, chromatograms and mass spectra were evaluated as previously described [51]”

Reference 51 does not exist

Author Response

Response to Comments

We appreciate the comments that the reviewers have given in our manuscript and the constructive criticism they have given. We have carefully reviewed the comments and have revised the manuscript accordingly. We believe that these changes have clearly improved our manuscript.

Response to Reviewer 1 Comments

  1. Overall, I see no major flaws in the transcriptome analysis. Likewise, I do not see major flaws in the GC-MS metabolite profiling. However, I also see no useful conclusions drawn from these analyses. In both cases a list of differentially accumulated transcripts or metabolites were found and no further conclusions were drawn. Similarly, a STRING protein-protein interaction network was generated but nothing was said about this. My biggest concern is that this manuscript has little or no impact. This manuscript cites Zeng et at. 2022 who have recently reported SGR1 knockout along with transcriptome analysis. It is not clear how this study adds knowledge or resources to the field.

>> Thank you for kindly reviewing the author.

In our manuscript, the SGR1 gene, which inhibits the expression of the PSY1 gene, which is most important for carotenoid biosynthesis, was generated in the sgr1 KO line by the CRISPR/Cas9 system. After selecting a null line among the sgr1 mutant, RNA sequencing analysis was performed. As a result, it was shown that the sgr1-KO null line significantly affected the expression of genes related to photosynthesis, chloroplast and carotenoid biosynthesis compared to the WT. Therefore, seeing strong changes in pigment and carotenoid content, other primary metabolic profiles were investigated. As a result, the leaves and fruits of the sgr1-KO null lines showed higher leveld of sucrose and its derivatives (e.g., fructose, galactinol, raffinose), its intermediates (e.g., glucose, G6P, Fru6P) and TCA cycle intermediates (e.g., malate and fumarate) compared to those of WT. Therefore, we believe that our manuscript has presented a very important conclusion in this respect. In addition, It was different from the paper published by Zeng et al. in 2022.

  1. The authors claim a null mutation.I think the sequencing data is convincing and shows a homozygous deletion (likely also a frameshift) mutation. Also, the phenotype is quite dramatic so I have no problem with claims that the induced mutation are effecting the phenotype. However, the authors did not conclusively prove a null mutation. It would be good to see RNA-seq quantification for SGR1 in the mutant line. This could show potentially decreased expression. More significantly, the read depth of these experiments may be sufficient to show a truncated transcript. This data may be enough to prove a null mutation. If not, the authors should not make this claim.

>> Thank you for kindly reviewing the author. The authors have inserted sentences in lines 111-141. Based on reviewer comments, Selection and Characterization of sgr1 Null Lines was described subheadings. Genetic segregation of the T-DNA region was calculated in the T1 generation using the sgr1#1-6 and sgr1#2-4 lines. In addition, as a result of resequencing analysis using these lines, it was proved that no segment was inserted from the outside into the genome of the sgr1-null lines.

  1. Overall, the manuscript was difficult to read for several reasons:

1)The manuscript would benefit from further editing. grammatical errors make it difficult to
understand certain passages (two examples below, but many exist in the text)

2)The introduction could be better organized. The basic carotenoid biosynthesis pathway is described and this would benefit from a diagram of the pathway.

3)Several different knockdown and knockout studies are cited in the intro but the text does
not bring the results of these studies together and it is not clear how the gene of interest (SGR1) relates to the existing knowledge cited.

>> Thank you for kindly reviewing the author. As pointed out by the reviewers, the authors performed English proofreading on Editage ([email protected]) to correct and supplement the manuscript.

  1. Two examples of grammatical errors that make the text unclear:

Line 52:“Overexpression of the lycopene β-cyclase 2 (LCY-B2) gene or the phytoene synthase 1 gene (PSY1) in transgenic plants have been increased lycopene levels in tomato fruits”

Maybe should read: “Overexpression of the lycopene β-cyclase 2 (LCY-B2) gene or the phytoene synthase 1 gene (PSY1) in transgenic plants has been show to increased lycopene levels in tomato fruits”.

Line 56 “In addition, the RNAi experiment of the green maintenance-related SGR1 gene, the contents of lycopene and β-carotene were accumulated 4 times and 9 times in fruit, respectively”

Maybe should read: “In addition, RNAi experiments have knocked down the green maintenance-related SGR1 gene. In these plants, lycopene and β-carotene in fruit accumulated to levels 4 and 9 times higher than wildtype, respectively”.

>> The authors fully agree with the points made by the reviewers. Therefore, our manuscript was replaced with the sentence suggested by the reviewer.

  1. Line 85: “and confirmed T0 transformants harboring target mutations for all two constructs using PCR-based genotyping (Figure 2A, Table 1).” This sentence confuses the reader. The phrase “for all two constructs” does not make sense. Also, the term PCR-based genotyping and referring to Fig2A seems to indicate that targeted mutations were confirmed via the gels in Fig2A. If I understand correctly, the authors do confirm target mutations but this is done through deep sequencing. This is not explained clearly in the results or methods section.

>> Thank you for kindly reviewing the author. As described by the reviewer, we have corrected the confusing sentence about target mutation by reorganizing it.

  1. Please clarify if the T0 or T1 lines (lacking the T-DNA) were used for all further analysis.

>> For further analysis, all analyzes were performed on null lines selected from the T1 generation.

  1. Line 131 “Chlorophyll content increased in sgr1 #1-6 and sgr1 #2-4 lines compared to WT (except for a slight decrease in chlorophyll b in sgr1 #2-4) (Table 3)” Table 3 shows a slight increase in IIb, not a decrease

>> It has been modified as follows: “Chlorophyll content increased in sgr1 #1-6 and sgr  #2-4 lines compared to WT (except for a slight increase in chlorophyll b in sgr #2-4) (Table 3)

  1. Line 148, “Principal component analysis of sgr1 #1-6 and WT libraries was used to determine data clustering based on SGR1 All biological replicates of sgr1 #1-6 and WT plants were distributed in two distinct groups (Figure 4A).” Figure 4A only seems to show one replicate from each group. This plot should show a point for each sample

>> Thank you for kindly reviewing the author. The authors made an error of representation in figure 4A. Therefore, reflecting the reviewer's opinion, it was rewritten including data from two repeated samples.

  1. GO enrichment analysis was carried out on DEGs. The authors should report details on how this analysis was carried out in the methods section.

>> As pointed out by the reviewer, details of GO enrichment analysis according to DEG were written in the method section.

  1. Figure 4 legend seems to indicate annotations for significance levels in the GO analysis. However, the figure contains none of these annotations.

>> As noted by the reviewers, the significance level of the GO analysis is inserted in Table 4.

  1. Table 5: It seems that all but one of the GO categories listed in this table are not significant (p-values > 0.05)

>> As noted by the reviewers, the significance level of the GO analysis is inserted in Table 5.

Minor comments

  1. Please double check references. As an example:“These results were similar to the fruit phenotypes generated in the SlSGR1-RNAi tomato line as well as in the SISGR1 knockout line using the CRISPR-Cas9 gene editing205 technique [14, 37, 39]”

It seems like this should be referencing [36], Zeng et al., 2022.

>> The authors thank the reviewers for their comments. All references have been checked and corrected.

  1. Line 301 Typo:“All raw reads are deposited at All raw reads are deposited at TeRaGene”

>> Edited according to the reviewer's comments.

  1. Line 325: “For metabolites, chromatograms and mass spectra were evaluated as previously described [51]” Reference 51 does not exist

>> The authors thank the reviewers for their comments. All references have been checked and corrected.

Reviewer 2 Report

The SGR1 (STAYGREEN1) protein is a critical regulator of plant leaves in chlorophyll degradation and senescence. In this paper, Kim et al. report on the transcriptome and metabolite profiles of SGR1-KO lines, and presented  new evidence for the mechanisms underlying the effects of SGR1 and molecular pathways involved in chlorophyll degradation and carotenoid biosynthesis.

Basically, the manuscript is well written and the science makes sense. The paper is free of major flaws. The authors don’t present any breathtaking news, but the involvement of SGR1 in chlorophyll degradation and senescence is still controversially discussed, and relevant contributions are welcome.

My major criticisms are:

  •       The cited literature in the introduction is kind of overwhelming. Out of 50 citations in the whole submitted ms, 37 are being found in the introduction. Wouldn’t it be enough to introduce or guide the reader to some seminal reviews? I don’t think that this means ‘ignoring’ other work. However, even for a specialized journal such as IJMS the introduction should try to address a broader readership.

  • What is the rational for distinction beween "results" and "supplemental results". I cant understand why some data are of supplemental value, and others not. Please explain.
  •  
  • Fig 4:  Differential gene expression analysis revealed 728 DEGs between WT and sgr 1-6 line, including 263 and 465 downregulated and upregulated genes, respectively, for which fold change was >2. Sounds good. However, did the authors make a test with PCR to validate some of the regulated genes? Principal component analysis and a volcano plot showing the DEGs between 166 WT and sgr1-mutation plants is great, but how reliable are the data?

Provided the authors address the concerns raised above I appreciate the paper. Distinction between main text/figures and supplements (Tables S1-7) seems to be quite odd. The presented research could provide a significant contribution to our understanding of chlorophyll degradation and carotenoid biosynthesis.

Author Response

Response to Comments

We appreciate the comments that the reviewers have given in our manuscript and the constructive criticism they have given. We have carefully reviewed the comments and have revised the manuscript accordingly. We believe that these changes have clearly improved our manuscript.

Response to Reviewer 2 Comments

My major criticisms are:

  1. The cited literature in the introduction is kind of overwhelming. Out of 50 citations in the whole submitted ms, 37 are being found in the introduction. Wouldn’t it be enough to introduce or guide the reader to some seminal reviews? I don’t think that this means ‘ignoring’ other work. However, even for a specialized journal such as IJMS the introduction should try to address a broader readership.

>> The number of references has been reduced to reflect the pointed comments of reviewers.

  1. What is the rational for distinction beween "results" and "supplemental results". I cant understand why some data are of supplemental value, and others not. Please explain.

>> As pointed out by the reviewers, the results presented in the results and supplementary materials were reviewed and corrected.

  1. Fig 4:  Differential gene expression analysis revealed 728 DEGs between WT and sgr 1-6 line, including 263 and 465 downregulated and upregulated genes, respectively, for which fold change was >2. Sounds good. However, did the authors make a test with PCR to validate some of the regulated genes? Principal component analysis and a volcano plot showing the DEGs between 166 WT and sgr1-mutation plants is great, but how reliable are the data?

>> Thanks for the reviewer's comments. Validation experiments were performed for the top genes showing DEG. The result is in Figure 5. In addition, the information obtained from our principal component analysis and volcano plot analysis appeared to center on genes involved in fruit maturation and carotenoid biosynthesis. Therefore, it was thought that there was no problem with the reliability of the data.

Reviewer 3 Report

The authors have investigated the function of the SGR1 gene in plants (tomato) with various well-established methodologies, including the genetic knockout based on CRISPR/Cas9-mediated gene editing. Besides the morphological (phenotypic) and biochemical variations in the knockout lines, the authors have identified the differentially expressed genes related to photosynthesis and chloroplast.

I believe that the authors have provided sufficient background and explained well the methodologies, though both GO and STRING analyses are missing from the M&M. However, I do have some major concerns. In particular, the authors should improve the overall presentation of the results and Discussion, as I have indicated below with more details. I believe that these revisions are fair and should significantly improve the overall presentation of this manuscript.

In many cases, the authors did not differentiate the gene from proteins, i.e., the common practice is to italicize the gene name but not the protein name. Please check and keep this format consistently throughout the entire manuscript. Also, some latin names are not italicized, e.g., line 61, Oryza sativa.

Line 68, to make this a new paragraph

Figures 1 and 2 each needs an overall title. Also, on Figures 1-3, the number “0” is given with a weird symbol, please correct it to “0”

Figure 1: the red arrows in part C need explanations.

Figure 2B: the alignment is clearly not correctly displayed…in particular #1-7, #1-8, and #1-9, as well as #2-6, #2-7, #2-8, and #2-9. No need to show the information of “-1bp” “-2bp” “-3bp” etc., because this information is straightforward on the alignments and it is not appropriate to use “deletion” or “insertion” for these indels.

Figure 3: please correct the grammatical errors in the overall title of this figure.

Line 126: change “was” to “were”

Tables 2 and 3: I would recommend that the data are presented as “mean ± SD” in one column, instead of two columns, and change the relevant wordings as well.

Table 5: the notes “*” at the bottom of the table contain incomplete information, “proc”???

Figure 4: part A presented the PCA of “two” samples but Table 4 contained 6 samples, are the “two” samples on PCA are the combinations of three WT and three mutation samples??? Also, in part C, instead of GO term numbers, it is more informative and appropriate to present the actual text descriptions of these GO terms, as in Table 5.

Figure 5: line 173, change “Genes” to “Gene”

Figure 6: the full name of “TCA” should be given somewhere, either on the figure or in the title of this figure.

Discussion: I am expecting to see more detailed discussion of the findings revealed in the current study in relation to the studies reported in the literature.

M&M: the GO annotation and STRING analyses are missing in the M&M but results of these two analyses are presented in Figure 4.

Author Response

Response to Comments

We appreciate the comments that the reviewers have given in our manuscript and the constructive criticism they have given. We have carefully reviewed the comments and have revised the manuscript accordingly. We believe that these changes have clearly improved our manuscript.

Response to Reviewer 3 Comments

  1. I believe that the authors have provided sufficient background and explained well the methodologies, though both GO and STRING analyses are missing from the M&M. However, I do have some major concerns. In particular, the authors should improve the overall presentation of the results and Discussion, as I have indicated below with more details. I believe that these revisions are fair and should significantly improve the overall presentation of this manuscript.

>> As pointed out by the reviewer, details of GO enrichment analysis according to DEG were written in the method section. In addition, The authors have revised and supplemented the manuscript as a whole.

  1. In many cases, the authors did not differentiate the gene from proteins, i.e., the common practice is to italicize the gene name but not the protein name. Please check and keep this format consistently throughout the entire manuscript. Also, some latin names are not italicized, e.g., line 61, Oryza sativa.

>> The author corrected and supplemented the italics of proteins and genes in the manuscript.

  1. Line 68, to make this a new paragraph

>> The points pointed out by the reviewer have been corrected.

  1. Figures 1 and 2 each needs an overall title. Also, on Figures 1-3, the number “0” is given with a weird symbol, please correct it to “0”

>> The points pointed out by the reviewer have been corrected.

  1. Figure 1: the red arrows in part C need explanations.

>> Explanted the arrows in Figure 1C pointed out by the reviewer.

  1. Figure 2B: the alignment is clearly not correctly displayed…in particular #1-7, #1-8, and #1-9, as well as #2-6, #2-7, #2-8, and #2-9. No need to show the information of “-1bp” “-2bp” “-3bp” etc., because this information is straightforward on the alignments and it is not appropriate to use “deletion” or “insertion” for these indels.

>>Edited based on the reviewer's comments.

  1. Figure 3: please correct the grammatical errors in the overall title of this figure.

>>Edited based on the reviewer's comments.

  1. .Line 126: change “was” to “were”

>>Edited based on the reviewer's comments.

  1. Tables 2 and 3: I would recommend that the data are presented as “mean ± SD” in one column, instead of two columns, and change the relevant wordings as well.

>>Edited based on the reviewer's comments.

  1. Table 5: the notes “*” at the bottom of the table contain incomplete information, “proc”???

>> Removed Proc, which is incomplete information from comments.

  1. Figure 4: part A presented the PCA of “two” samples but Table 4 contained 6 samples, are the “two” samples on PCA are the combinations of three WT and three mutation samples??? Also, in part C, instead of GO term numbers, it is more informative and appropriate to present the actual text descriptions of these GO terms, as in Table 5.

>>As the reviewer pointed out, Figure 4A re-labeled the 6-sample data. In Figure 4C, gene numbers are re-labeled instead of GO terms.

  1. Figure 5: line 173, change “Genes” to “Gene”

>>Edited based on the reviewer's comments.

  1. Figure 6: the full name of “TCA” should be given somewhere, either on the figure or in the title of this figure.

>> The authors changed the title of Figure 6 to "Metabolite profiling in sucrosec and its derivatives, glycolytic intermediates and TCA cycle intermedi-ates in the leaves and fruits by sgr1 knockout in tomato".

  1. Discussion: I am expecting to see more detailed discussion of the findings revealed in the current study in relation to the studies reported in the literature.

>> It was corrected and supplemented based on the reviewer's detailed review.

  1. M&M: the GO annotation and STRING analyses are missing in the M&M but results of these two analyses are presented in Figure 4.

>> GO annotations and STRING analysis are described in M&M.

Round 2

Reviewer 3 Report

I appreciate very much the efforts that the authors have devoted to improving their manuscript. I have no more questions.